# Dead cells release a 'necrosignal' that activates antibiotic survival pathways in bacterial swarms

Souvik Bhattacharyya [1], David M. Walker [1] & Rasika M. Harshey [1]✉

Swarming is a form of collective bacterial motion enabled by flagella on the surface of semi-solid media. Swarming populations exhibit non-genetic or adaptive resistance to antibiotics, despite sustaining considerable cell death. Here, we show that antibiotic-induced death of a sub-population benefits the swarm by enhancing adaptive resistance in the surviving cells. Killed cells release a resistance-enhancing factor that we identify as AcrA, a periplasmic component of RND efflux pumps. The released AcrA interacts on the surface of live cells with an outer membrane component of the efflux pump, TolC, stimulating drug efflux and inducing expression of other efflux pumps. This phenomenon, which we call 'necrosignaling', exists in other Gram-negative and Gram-positive bacteria and displays species-specificity. Given that adaptive resistance is a known incubator for evolving genetic resistance, our findings might be clinically relevant to the rise of multidrug resistance.

[1] Department of Molecular Biosciences, University of Texas at Austin, Austin, TX 78712, USA. ✉email: rasika@austin.utexas.edu

Bacteria employ many appendages for movement and dispersal in their ecological niches[1]. Of these, flagella-driven motility is the fastest, promoting the colonization of favorable niches in response to environmental signals, and contributing significantly to the pathogenic ability of some species[2–5]. Flagella are used both for swimming individually through liquid, and swarming collectively over surfaces[3,6,7]. An unexpected and clinically relevant property of swarms is their nongenetic resistance to antibiotics at levels lethal to free-swimming planktonic cells of the same species[8–12]. The resistance depends on high cell densities associated with swarms[12], and is phenomenologically similar to adaptive resistance of bacterial biofilms[13]. While several terminologies such as resistance, tolerance, and persistence have been used to describe bacterial resistance phenotypes observed under various settings[14,15], a shared attribute of a majority these phenotypes is their association with reduced metabolism and slow bacterial growth[16]. In stark contrast, bacterial swarms are metabolically active and grow robustly, suggesting that different resistance mechanism/s may be operative here. To distinguish the adaptive resistance seen in swarms from that in other reported cases of this phenomenon, we are calling swarming-specific resistance SR.

In a study of several bacterial species, swarms that successfully colonized media containing antibiotics nonetheless experienced substantial cell death[12]. Programmed bacterial cell death, where death of a subpopulation benefits the community by providing nutrients, has been reported by several studies[17–19]. Cell death incurred during swarming, however, appeared to provide a protective benefit, leading to speculation that a dead-cell layer at the bottom of the swarm might shield live cells swarming on top by acting as a physical barrier against antibiotic penetration[12]. In a study examining the 3D architecture of a swarm in the presence of different antibiotics, dead cells were not always found at the bottom, ruling out a barrier role[20]. Instead, live cells were seen to migrate away from the antibiotic source, leading to a suggestion that cell death emanating around this source releases an avoidance "signal" that contributes to the resistance[20].

The impetus for the present study came from our observation of heterogeneity in antibiotic-induced cell killing in a swarm, where a distinct subpopulation was more susceptible to killing. To test if death of this subpopulation contributes to SR, pre-killed cells were introduced into the swarm, where they indeed enhanced SR. In a study focused on *Escherichia coli* (*E. coli*), we have used this observation to purify the SR-enhancing factor. We show that the dead cells release a "necrosignal" in the form of AcrA, which is a periplasmic protein component of a tripartite RND efflux pump. AcrA binds to the TolC component of this pump on the outer membrane (OM) of live cells to activate immediate efflux, as well as stimulates expression of multiple pathways that overcome antibiotic stress, pathways that are already enhanced in the swarm. Our finding of heterogeneity in cell killing within the swarm is suggestive of altruistic features that ultimately increase survival of the swarm. Finally, we demonstrate that necrosignaling operates in multiple bacterial species and exhibits species specificity.

## Results

### Death of a subpopulation may be linked to SR.
Initially, we monitored the antibiotic susceptibility of planktonic vs. swarm cells of *E. coli*. Exposure to a broad concentration range of the antibiotic kanamycin (Kan) produced distinct survival patterns (Supplementary Fig. 1). At low concentrations (e.g, Kan[2.5]; 2.5 μg/ml), and during the early phase of growth (0.5 h), swarm cells were surprisingly more susceptible to killing than planktonic cells, a trend maintained for up to Kan[10] (Supplementary Fig. 1a, b). This trend

reversed at both higher Kan concentrations as well as later times, where planktonic cells became more susceptible. We hypothesized that the higher initial susceptibility of swarm cells could be indicative of heterogeneity in the population, with a subpopulation that was more sensitive. This rationale was tested in silico by simulating the survival patterns of each subpopulation as a first-order decay where the survival "rate" is analogous to that population's susceptibility to the antibiotic (Supplementary Fig. 1a and "Methods"). A homogeneous cell population with a singular response rate reasonably modeled the planktonic data (MSE range: $10^{-6}$–$10^{-3}$), but failed to model the swarm data (MSE range: $10^{-3}$–$10^{-2}$) (Fig. 1a). Conversely, when a heterogeneous population composed of two subpopulations differing in initial rates of death were introduced into the simulations, the resultant curves successfully fit the experimental data as determined by MSE scores 10–1000 times lower (MSE range: $10^{-7}$–$10^{-5}$) compared to the homogeneous population. The fits between the two population models for both swarming and planktonic cells were determined to be statistically significant by the Wilcoxon rank sum test (Fig. 1a). Taken together, the data indicate the existence of two distinct subpopulations within the swarm, one more susceptible to the antibiotic. Given that the experimental MIC for Kan in swarm cells was almost double that of planktonic cells (Supplementary Fig. 1c), we hypothesized that the more susceptible population might serve as an early distress signal to elicit resistance in the swarm community.

### Dead cells release AcrA as a necrosignal.
To directly test whether cell death acts as a signal, the border-crossing assay[12] was primarily employed as diagrammed in Fig. 2a, where in a divided petri plate containing swarm media, only the right chamber has the antibiotic. To test if cell death enhances SR, pre-killed cells were applied to the right chamber, now containing antibiotic at a concentration higher than the swarm's normal tolerance (Fig. 2b). WT *E. coli* inoculated on the left, swarm over the right chamber with Kan[25] (not shown) but not with Kan[50] (Fig. 1 b1). When *E. coli* cells killed by Kan[250] were applied to the right chamber, the WT population could colonize Kan[50] (Fig. 1 b2). Although cells killed with Kan promoted migration over Kan[50], the enhanced resistance was independent of the killing method (Supplementary Fig. 2a), with the exception of heat (Supplementary Fig. 2b). The response to killed cells was sustained, in that the swarm retained its capacity for resistance even after exiting a zone of dead cells (Supplementary Fig. 3).

The heat-sensitive nature of the SR-factor (Supplementary Fig. 2b) suggested that it might be isolatable. To this end, cell extracts prepared from Kan[250]-treated cells were assayed, and showed activity in the supernatant fraction (Supplementary Fig. 2b). The activity was resistant to DNaseI and RNaseI (Supplementary Fig. 2c), but sensitive to protease (Fig. 1 b3). A 30% ammonium sulfate precipitate, when resuspended in buffer and applied as lines, promoted the swarm to track along these lines (Fig. 1 b4). We will henceforth refer to this active factor as the necrosignal, and its ability to promote SR as "necrosignaling".

We found necrosignaling to be operative in other bacterial species as well (Fig. 1e, Supplementary Fig. 2d). However, except for *E. coli* and *Salmonella*, where killed cells of one species promoted a reciprocal response in the other (Fig. 1e, maroon areas), species-specificity of the response was evident in *Bacillus subtilis*, *Pseudomonas aeruginosa*, and *Serratia marcescens* (Fig. 1e, blue areas). Given that *E. coli* and *Salmonella* have an interchangeable response, we used both bacteria to purify and determine the common identity of the necrosignal (Supplementary Fig. 4). MS/MS analysis of the active fractions obtained after the final purification step yielded five common proteins (Supplementary Fig. 5; AcrA, UspE, BaeR, YhdC, and Crp). All subsequent experiments were performed with *E. coli*.

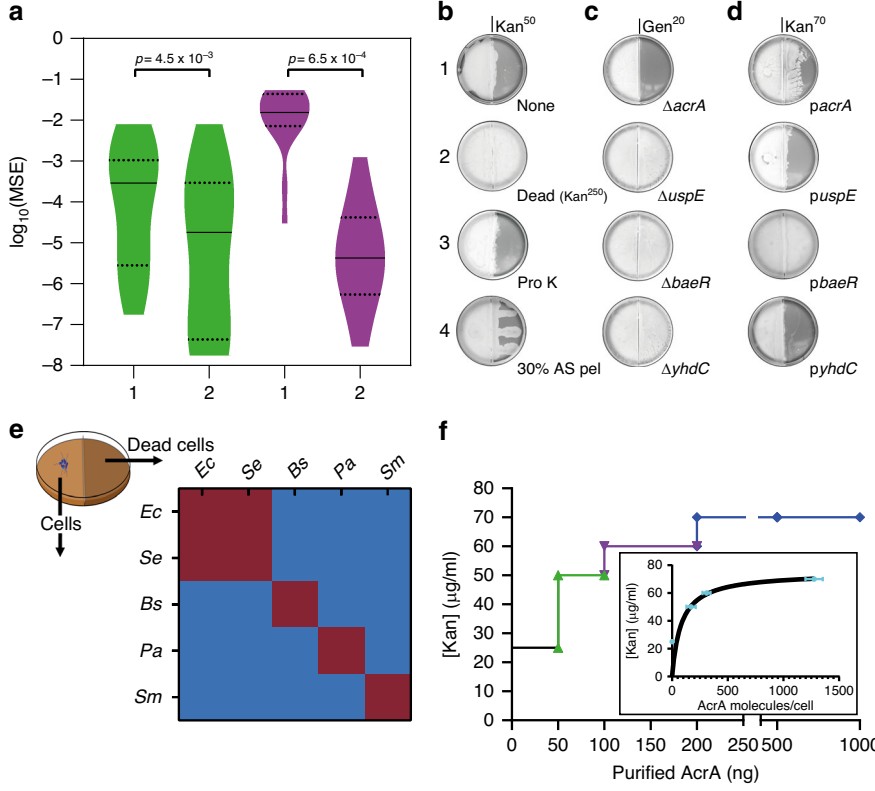

**Fig. 1 Dead bacteria release a necrosignal. a** Graph showing distribution of the mean squared errors (MSE) between experimental and simulated killing curves of planktonic (green) and swarm (purple) cells shown in Supplementary Fig. 1a; 1 and 2 are simulations assuming homogeneous and heterogeneous populations, respectively (see "Methods"). The indicated $p$ values were calculated from a two-tailed Wilcoxon rank sum test between the two types of populations. Median, solid black lines; quartiles, dashed black lines. **b–d** Border-crossing assays that established the identity of the necrosignal. WT *E. coli* were inoculated in the left chamber in every case, whereas material applied to the right chamber is indicated below each plate. **b** None, no cells applied; Dead (Kan$^{250}$), cells killed by Kan$^{250}$; Pro K, cell extract supernatant from killed cells, treated with Proteinase K (see Supplementary Fig. 2b for supernatant alone); AS pel pellet fraction after treating supernatant with ammonium sulfate. Kan Kanamycin, Gen Gentamycin. **c** Gene deletions ($\Delta$). All gene deletions were made with a Kan cassette, so the cells were pre-killed with Gentamycin (Gen$^{50}$), and tested for swarming on Gen$^{20}$. b2 serves as the control for these experiments. **d** Gene overexpression from ASKA library plasmids (p). These strains were pre-killed with Kan$^{250}$. **e** Chart showing the species specificity of necrosignaling. *Ec Escherichia coli, Se Salmonella enterica, Bs Bacillus subtilis, Pa Pseudomonas aeruginosa, Sm Serratia marcescens.* Columns: bacterial species inoculated in the left chamber. Rows: bacterial species providing the dead cells applied on the right chamber, Maroon, SR+ response; Blue, SR− response. See "Methods" for the cell killing procedure and assay conditions, and Supplementary Fig. 2d for the raw data. **f** Swarming response to indicated Kan concentrations with increasing AcrA applied to the right chamber. The response was saturated at Kan$^{70}$. Inset: minimum number of AcrA molecules estimated to be required for SR (calculated from moles of AcrA required per CFU/ml of swarm cells, $n = 3$) at the indicated Kan concentrations. The data (cyan dots) were fit to an exponential function (black line). Data are presented as mean values ± SD. Source data are provided as a Source Data file.

Deletion and overexpression analysis helped narrow down the necrosignal candidate (Fig. 1c, d). In both analyses, candidate strains were pre-killed and applied to the right chamber. Only $\Delta acrA$ abolished the enhanced resistance response to Kan$^{50}$ (Fig. 1c; $\Delta crp$ is not included because this deletion severely represses swarming[21]). Conversely, when overexpressed, AcrA (p$acrA$) promoted swarming over Kan$^{70}$, but so did BaeR (p$baeR$) (Fig. 1d). BaeR is a positive regulator of $acrAD$ operon[22]. Given that $\Delta baeR$ did not abolish the response, AcrA is most likely the necrosignal. Purified AcrA (flanked by His- and FLAG-epitope tags; see "Methods" and Supplementary Fig. 4d), showed a concentration-dependent SR response, plateauing at Kan$^{70}$ (Fig. 1f). The estimated number of AcrA molecules per cell required to elicit a response increased exponentially with increasing antibiotic (Fig. 1f, inset) reaching a plateau, suggesting specificity (nonspecific binding is generally linear[23]).

When purified AcrA was added to a planktonic culture treated with Kan near its MIC$_{99}$ (see Supplementary Fig. 1c for MIC analysis), only a modest increase in survival was observed, unless cell density was increased (Fig. 3). Even at high planktonic cell density, swarm cells had a better survival response, indicating that high cell density is important but not sufficient to account for swarm resistance, and that the physiological state of the swarm might be an important contributor as well.

**Mechanism of AcrA necrosignaling.** AcrA is the periplasmic component of the RND efflux pump AcrAB-TolC (and AcrAD-TolC)[24]. To elucidate whether the activity of this surprising candidate for the necrosignal was independent of the OM component TolC, we carried out both deletion and complementation analysis with *tolC*. Absence of TolC in the active swarm annulled the resistance response to AcrA applied on the right (Fig. 4 a1), while complementation with TolC restored it (Fig. 4 a2). However, absence of TolC in killed cells applied on the right did not affect the response outcome (Fig. 4 a3). Taken together, these results indicate that the necrosignaling activity of AcrA is dependent on the presence of TolC in the live cells, and suggest that AcrA released from dead cells may bind to TolC from the outside.

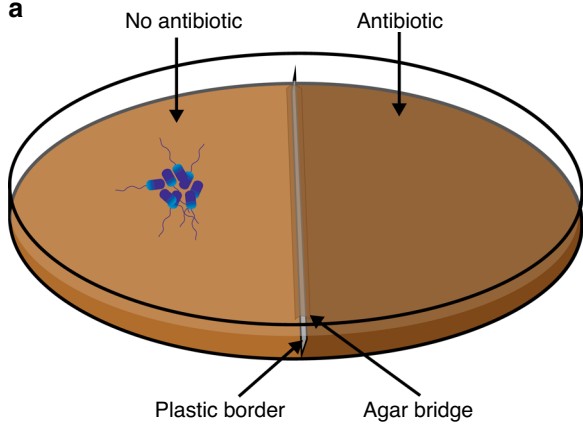

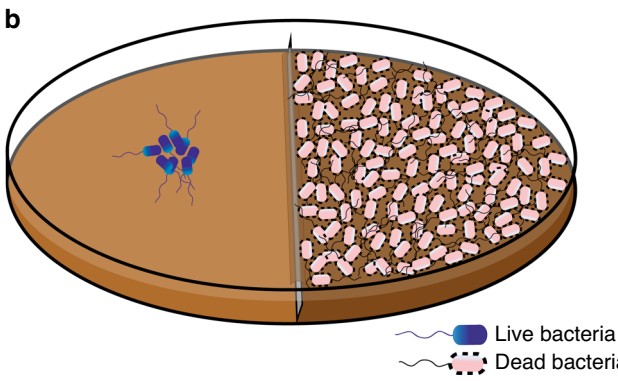

**Fig. 2 Border-crossing assay. a** Petri plates with a plastic divider create two chambers. The left chamber is poured with media without antibiotic, and the right chamber with antibiotic. After the media is set, the two chambers are connected by a thin layer of agar on the top of the bridge[12]. Bacteria are inoculated in the left chamber as indicated, and allowed to swarm to the right chamber. **b** As in (**a**), but with dead bacteria layered on the surface of media on right.

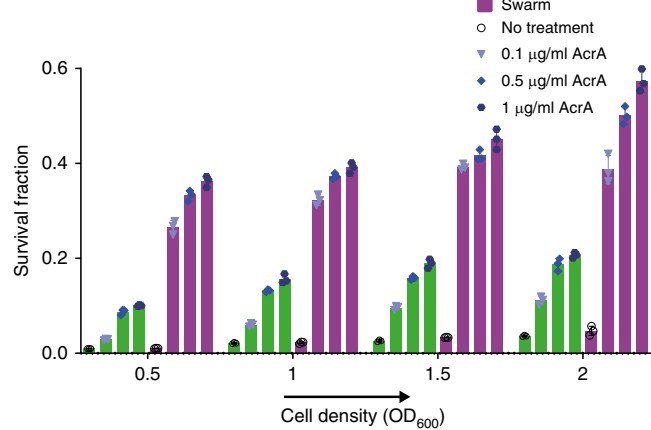

**Fig. 3 Necrosignaling activity of purified *E. coli* AcrA.** Mid-log phase planktonic (green) and swarm (purple) cells of *E. coli* at increasing densities ($OD_{600}$, *x* axis), either concentrated or diluted to achieve desired $OD_{600}$, were treated with Kan[20] ($MIC_{99}P$) and Kan[37.5] (~$MIC_{99}S$), respectively, for 1 h at 37 °C with or without the addition of purified AcrA at indicated concentrations. CFU counts of survivors were used to calculate the survival fractions. Each of the three individual replicates are shown on a single bar. The data were analyzed using a mixed model ANOVA. Cell type, i.e., planktonic and swarm cells, were taken as the "between-subjects" factor and a Giesser–Greenhouse correction was applied to it; cell density and AcrA treatment were considered as "repeated measures". The multiple comparisons were corrected using Dunnett testing, keeping a two-tailed significance level of 0.05. The obtained *p* value (0.000074) of interaction was significant. The maximum increase in survival observed was: (i) at 0.5 $OD_{600}$, ~9% in planktonic cells compared to ~25% in swarm cells and (ii) at 2.0 $OD_{600}$, ~17% in planktonic cells compared to ~50% in swarm cells. Source data are provided as a Source Data file.

We next sought to identify residues critical for the response in both AcrA and TolC. For TolC, we took advantage of reports that TolC serves as a receptor for certain colicins[25] and bacteriophages[26,27], using residues important for phage binding[26] as a guide for our mutagenesis (Fig. 4b). As shown in Fig. 4c, a subset of the residues tested were also essential for the SR response. For AcrA (Fig. 4b), deletion of 72 and 75 residues from its C- and N- terminus respectively, did not affect its activity, but perturbation of its helix-turn-helix (HTH) motif or nearby residues eliminated it (Fig. 4c), suggesting that the HTH region of AcrA is critical. (Primary data for results summarized in Fig. 4c are provided in Supplementary Fig. 6).

To test binding of AcrA to TolC exposed on the OM, we marked AcrA with Qdot[705]-labeled anti-FLAG antibody, and labeled the membrane with FM 1-43. Microscopy images show localization of AcrA to the membrane (Fig. 4d and Supplementary Fig. 7a). Deletion and complementation of *tolC* led to loss and restoration of AcrA binding, respectively (Fig. 4e and Supplementary Fig. 7b). Critical TolC residues as deduced from plate assays (Fig. 4c), were verified in the binding assay by the observation that *tolC*[S257A] but not *tolC*[R55A] supported AcrA binding (Fig. 4e and Supplementary Fig. 7b). The extracellular localization of AcrA was confirmed by trypsin digestion (Fig. 4e and Supplementary Fig. 7c; see Supplementary Fig. 7d for Qdot[705] control). Taken together, both genetics and microscopy corroborate that AcrA binds to TolC externally to elicit necrosignaling.

**Mechanism of SR**. To determine if necrosignaling impacts gene expression, RNA-seq analysis was performed. Expression of 566 genes, many of which ontologically classified to motility, membrane function, and energy metabolism, showed specific changes in swarm cells (Fig. 5a and Supplementary Fig. 8). Efflux pumps and transport functions were upregulated, with further increases in expression upon addition of antibiotics and of AcrA. Expression of OM porin genes were down in the swarm (Fig. 5a) and those of ROS catabolism genes were up (Fig. 5a). These results were verified in several ways.

The Nile Red assay for efflux[28,29] confirmed a higher rate of efflux, stimulated by AcrA (Fig. 5b). The Alamar Blue assay for membrane permeability[29,30] showed lower permeability, unaffected by AcrA (Fig. 5c). Next, we directly measured the intracellular concentration of antibiotics using two methods—spectrometry employing the fluorescent antibiotic Fleroxacin[31], and disc-diffusion assay for the thermostable Kanamycin[32]. The concentration of both antibiotics was significantly lower in swarm cells compared to planktonic cells (Fig. 5d, e), consistent with the data in Fig. 5b, c.

The RNA-seq data were further validated by genetics. Overexpression of selected genes involved in ROS catabolism, iron transport, and efflux pumps increased SR, whereas overexpression of porins decreased it (Supplementary Fig. 9a, c). Deletions of many efflux pump components and their regulators also decreased SR (Supplementary Fig. 9b, d).

Taken together, the data show that *E. coli* swarms are preprogrammed to be less permeable, more prepared to efflux, and to tolerate antibiotic stress by upregulating ROS catabolism pathways. Antibiotic exposure and the resultant cell death

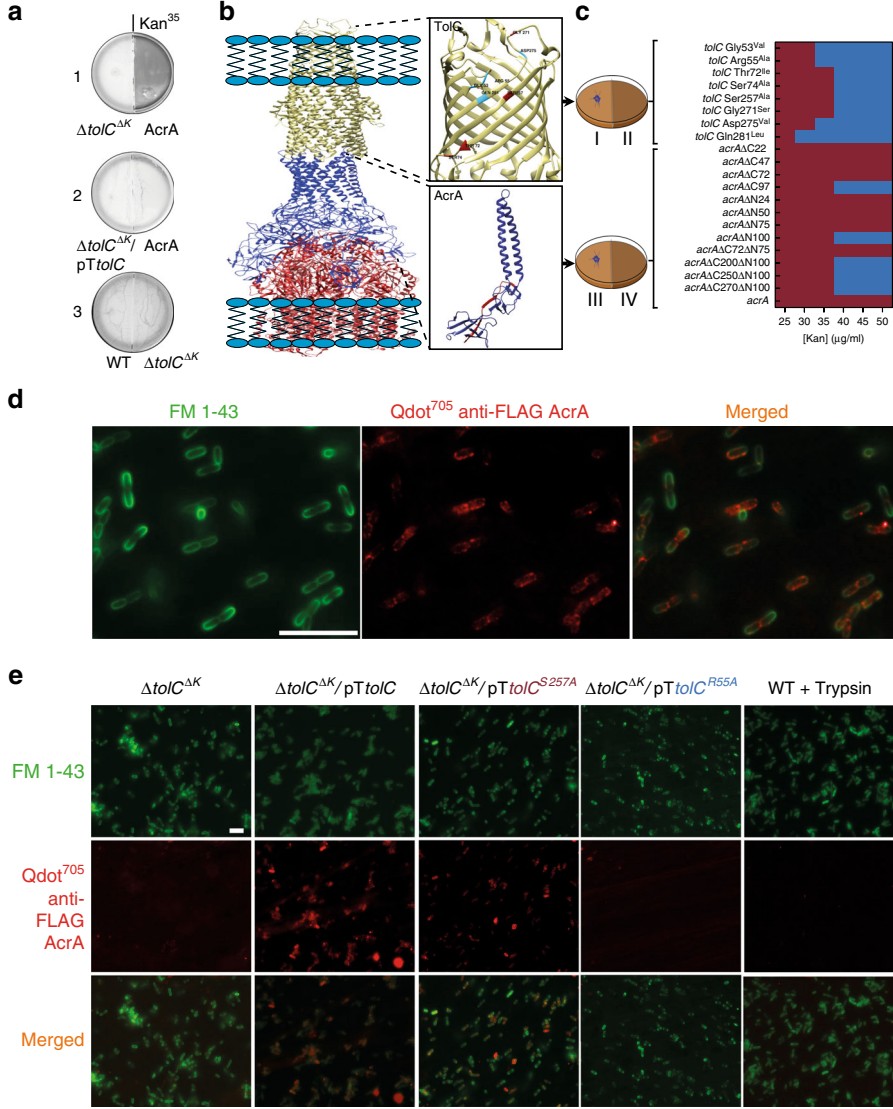

**Fig. 4 AcrA binds TolC externally as a necrosignal. a** The AcrA response requires TolC. Genotypes of *E. coli* strains inoculated on the left or proteins/pre-killed cells applied on the right are indicated below the plates. The *tolC* deletion reduces SR to Kan[25], therefore Kan[35] was used in these plates. Strain ΔtolC[ΔK] has its Kan[R] marker removed (ΔK; see Supplementary Table 1 and "Methods"). **b** A model of the AcrAB-TolC efflux pump (PDB ID: 5NG5) visualized and drawn (not to scale) using Chimera[93], with TolC and AcrA components enlarged. The enlarged view for TolC shows the residues mutated in this study, with maroon and blue indicating SR+ and SR− outcomes, and similar colors in AcrA indicating the importance of the HTH region for SR+ activity. **c** Summary of data delineating residues in TolC, and regions in AcrA, important for SR. For TolC analysis, *tolC* mutants expressed from pTrC99a plasmids in a ΔtolC[ΔK] strain were inoculated on the left (I), with purified AcrA on the right (II). For AcrA, WT cells were inoculated on the left (III), and pre-killed cells expressing the indicated C- and N-terminal truncated versions of AcrA from pTrc plasmids in a ΔacrA[ΔK] strain (Kan marker removed) were applied on right (IV). SR response color scheme as in Fig. 1e. See Supplementary Fig. 6 for primary data. **d** Imaging of AcrA binding to the outer membrane of WT swarm cells. QDot[705]-labeled anti-FLAG antibody was used to label AcrA-FLAG, and FM-143 dye to label the outer membrane (see "Methods"). The entire field of view is shown in Supplementary Fig. 7a. Scale bar, 10 μm. See Supplementary Fig. 7d for binding of Qdot alone (control). **e** AcrA-TolC co-localization in representative SR+ and SR− TolC mutants identified in (**c**) and treated as in (**d**). See Supplementary Fig. 7b for brightfield images. Scale bar, 10 μm. The last panel is a control to show that AcrA is not internalized. Here, WT swarm cells were pre-incubated with AcrA and subsequently treated with trypsin followed by addition of the Qdot (see "Methods"). See Supplementary Fig. 7c for western blot analysis of these samples. Source data are provided as a Source Data file.

and release of AcrA enhances these pathways further. AcrA also stimulates immediate efflux from TolC pumps. Figure 6 summarizes these results.

## Discussion

Bacterial swarms alter the expression of genes controlling both morphology and physiology[10,33–38]. We show here that *E. coli* swarms are intrinsically programmed to survive encounters with

antibiotics by modulating at least three different pathways as proactive shields: efflux, ROS catabolism, and membrane permeability. The first two pathways are upregulated, while the third is downregulated. The upregulation of multiple efflux pump components has also been reported in *Pseudomonas*[10,39] and *Salmonella*[35]. ROS stress is known to be generated by antibiotics[40–42], high cell densities[43,44], and increased energy utilization[45], the latter two being hallmarks of swarms[12,21,46]. Densely packed and highly motile swarm cells utilize more energy[46] via

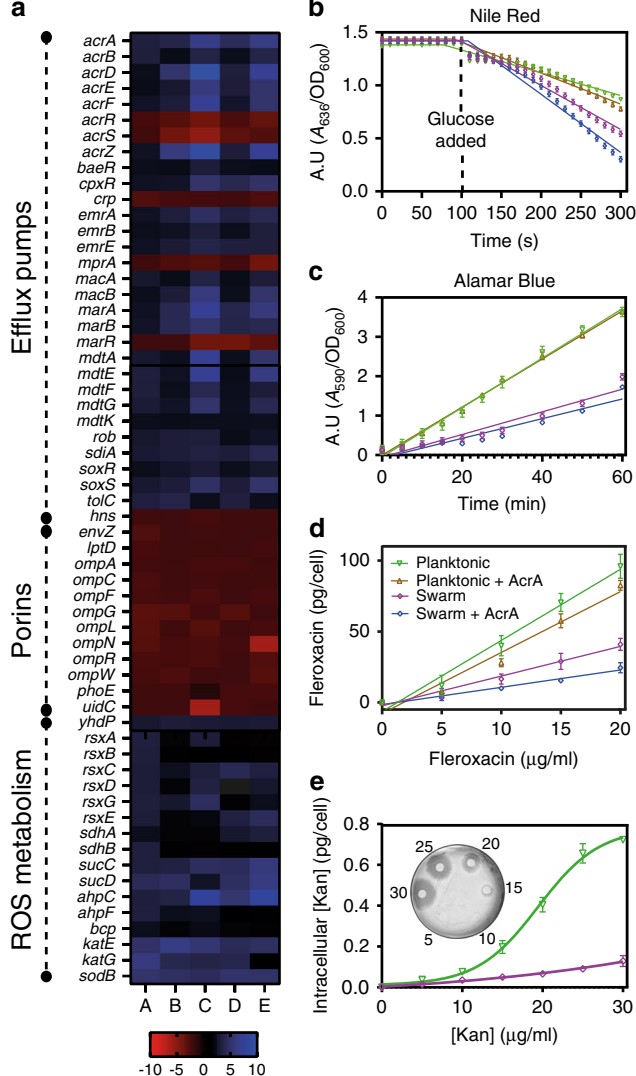

**Fig. 5 Mechanism of SR. a** Comparison of $\log_2$ fold changes in gene expression. (A) Swarm vs. Planktonic, (B) Swarm + Kan[20] vs. Swarm, (C) Swarm + Kan[20] + AcrA vs. Swarm, (D) Swarm + Cip[2.5] vs. Swarm, (E) Swarm + Cip[2.5] + AcrA vs. Swarm. The concentration of AcrA was 0.1 μg/ml, and the $\log_2$ fold change cutoff value was 2. About 25 genes encoding efflux pump components representing all five classes of pumps and their regulators were upregulated in swarm cells (~2–3 fold), while ten genes expressing different porins or their regulators were downregulated (~2–3 fold). Except for porins, all these genes were further upregulated (~2–6 fold) in swarm cells under antibiotic stress (Kan and Cip). The presence of AcrA also increased expression of these same genes (~2–9 fold). Twenty-three genes for energy metabolism (~2–4 fold) and 16 genes related to ROS catabolism (~2–6 fold) were also upregulated. **b** Efflux assay using Nile red indicator dye ($n = 3$). Glucose was added at 100 s. Nonlinear regression analysis of Planktonic, Planktonic + AcrA, Swarm, and Swarm + AcrA data sets (see **d** for color key) yielded rate constants of $1.001 \times 10^{-0.6}$, $8.669 \times 10^{-0.7}$, $6.630 \times 10^{-0.6}$, and $1.165 \times 10^{-0.6}$, respectively. A.U. = ($A_{636}$/$OD_{600}$). The time required for 50% Nile Red efflux or $t_{eff50}$, was significantly low in swarm cells (230 s for swarm and ≥300 s for planktonic cells; compare purple vs. green lines; $p < 0.0001$ for the two values). Addition of AcrA yielded $t_{efflux50}$ with a 1/10th increase in swarm cells (207 s, $p = 0.0003$ for Swarm + AcrA vs. Swarm from a two-tailed $T$ test) and 1/20th increase in planktonic cells (282 s, compare brown vs. blue lines; $p = 0.0038$ for planktonic +AcrA vs. Planktonic from a two-tailed $T$ test). **c** Membrane permeability assay using Alamar Blue indicator dye ($n = 3$). Nonlinear regression analysis of Swarm, Swarm + AcrA, Swim, and Swim+ AcrA data sets (see **d** for color key) produced slopes of 0.06254, 0.06103, 0.02876, and 0.02498 respectively. A.U. = ($A_{590}$/$OD_{600}$). **d** Estimated intracellular concentration of Fleroxacin ($n = 3$). The slopes of the fitted lines of data sets for Swarm, Swarm + AcrA, Planktonic, and Planktonic + AcrA were 2.090, 1.220, 5.010, and 4.290, respectively. **e** Estimation of intracellular antibiotic concentration in *E. coli* using the Disc-diffusion assay. See "Methods" for details. The inset image shows one representative plate with discs containing planktonic samples incubated with the Kan concentrations indicated by the numbers. Inhibition zones around the filter discs were compared to those from a standard curve generated with known concentrations of kanamycin. Swarm cells show ~8-fold decrease in intracellular [Kan] at Kan[30] compared to planktonic cells. Data are presented as mean values ± SD in (**b–e**). Source data are provided as a Source Data file.

extensive utilization of TCA cycle[21,46], which leads to intracellular ROS stress via Fenton reactions through agents such as $FADH_2$ and menaquinone[47]. Superoxide dismutase (SodB) and catalases (KatE, KatG) (Fig. 5a) protect the cells from damage to the Fe–S clusters, which are co-factors/prosthetic groups of various proteins[48]. The importance of iron homeostasis during swarming[10,21,35,36,49,50], essential for the regeneration Fe–S clusters of ROS catabolizing enzymes[51], was corroborated in our study as seen by the upregulation of multiple genes involved in uptake (*feoAB, fecABCDE, fhuABCD*; see Supplementary Fig. 8c) and metabolism of iron (*ahpC, rsxABCDE, sdhA*; see Fig. 5a). A decrease in membrane permeability has also been reported for *Salmonella* and *Proteus* swarmers[36,46].

Our study shows that despite encoding an intrinsic program for surviving antimicrobials, swarms deploy an added security measure by exploiting the death of a subpopulation (induced by antibiotic or other insults; Fig. 1b and Supplementary Fig. 2a) to communicate a state of emergency to the group to further activate efflux and ROS pathways. The novel mechanism by which cell death enhances tolerance to antibiotics is by the release of AcrA, a periplasmic constituent of TolC efflux pumps, which binds to TolC on live cells from the outside (Fig. 4d). AcrA interaction with TolC immediately stimulates efflux in the population (Fig. 5b) and upregulates the expression of a multitude of genes for a sustained SR response (Fig. 5a). How AcrA triggers these

events, and whether AcrA is also the signal that induces an avoidance response to antibiotics in the swarm[20], will be avenues of future research.

A comparison of AcrA from the bacterial species used in this study shows a high degree of sequence conservation (Supplementary Fig. 10a, b), the *E. coli* and *Salmonella* proteins being most closely related (Supplementary Fig. 10c), explaining their interchangeable necrosignaling response (Fig. 1e). Interestingly, TolC is absent in many bacteria including *Bacillus*. In contrast, AcrA is highly conserved across the Bacterial domain (Supplementary Fig. 11) suggesting it might have other binding partners, i.e., more necrosignaling modules may be found in nature. The existence of a specific signal-receptor survival module (AcrA: TolC) is expected to benefit its own species in a multispecies ecological community (Fig. 1e).

The surprising aspect of our findings is that a normal component of an RND drug efflux pump, AcrA, moonlights as a necrosignal directed at similar pumps. Another curious finding is the structural homology of AcrA to colicin E3 (Fig. 7a–c, RMSD = 0.85 Å), which interacts with BtuB (vitamin B12 receptor), a β-barrel OM protein similar to TolC[52,53]. We have used the colicin homology to suggest possible modes of AcrA binding to TolC (Fig. 7d, e). The best fit has striking similarities

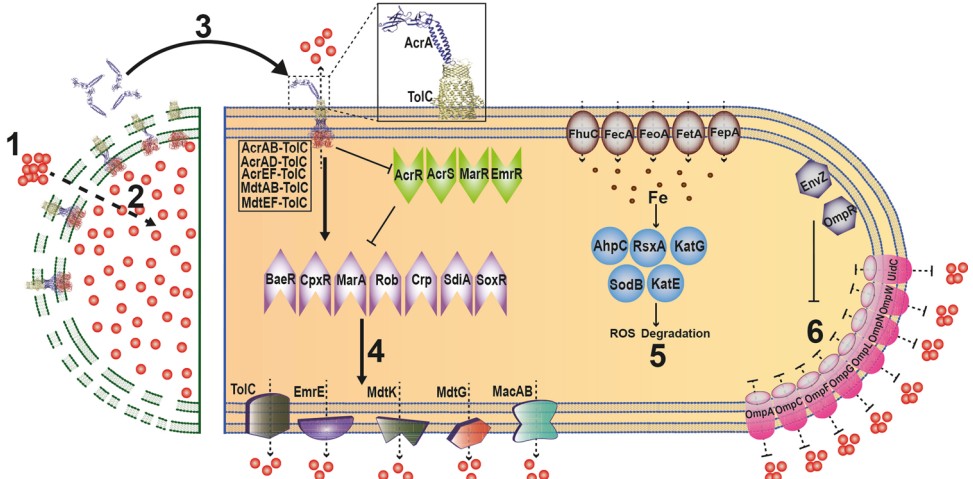

**Fig. 6 Model showing how necrosignaling promotes SR.** Both live and dead cells are represented in a single cartoon. **1** Antibiotic uptake (red dots). **2** Cell death, membrane damage. **3** Released AcrA binds to TolC in the OM of live cells, activating efflux from TolC pumps. **4** A secondary consequence of TolC activation is upregulation of five categories of efflux pumps. This upregulation is enabled by mulitple activators (purple), or by repression of multiple repressors (green). **5** Upregulation (arrowhead) of genes that reduce ROS (blue and brown) would allow tolerance of the antibiotic stress. **6** Downregulation (flathead) of porins (purple) would restrict entry of antibiotics.

with colicin E3–BtuB interaction[54], suggesting perhaps a case of molecular mimicry. While we do not know whether colicins and AcrA share a common evolutionary history, we speculate that such mimicry may have evolved in environmental niches where release of AcrA via colicin-induced cell death resulted in a competition for extracellular binding to TolC on live cells.

Death of a bacterial subpopulation for the overall survival of community and its resemblance to altruism, has been the subject of earlier studies[19,55,56]. While a variety of functions for cell lysis have been reported—recycling nutrients during bacterial starvation[57], exchange of genetic material[58], and activation of systems that deliver toxins to confer fitness in inter-species co-cultures[59]—the cell death response we report here has several features not shown in earlier studies. For example, the existence of a heterogeneous population with respect to antibiotic susceptibility (Fig. 1a and Supplementary Fig. 1a), and the advantage it confers, is akin to a bet-hedging survival strategy[19]. High cell densities provide another selective advantage given that a significant fraction of the swarm must be killed for SR to be observed (~1/2 at Kan[25]; Supplementary Fig. 1a), an observation that should be considered for clinical antibiotic regimen guidelines[60] and in ecological studies involving microbial interactions[61,62]. A large motile population, that can continuously acquire new territory, and that has a strategy to overcome stressful environments and survive mass extinction events promoted by the event itself, will also increase its chances of propagating beneficial mutations such as antibiotic resistance. We note that a response to cell death that benefits the living is not confined to bacteria, but is widespread in nature, as seen in insects (necromones)[63,64], fishes[65], birds[66], and mammals[67], possibly even in eukaryotic tumors[68].

## Methods

**Strains, plasmids, and growth conditions.** Bacterial strains, plasmids, and primers used in this study are listed in Supplementary Table 1. The strains were propagated in LB (10 g/L tryptone, 5 g/L yeast extract, and 5 g/L NaCl) broth or on 1.5% Bacto agar plates for single colony isolation. Antibiotics for marker selection were added as follows: Kan (Kanamycin) 25 μg/ml, Amp (Ampicillin) 100 μg/ml, and Cam (Chloramphenicol) 30 μg/ml. Antibiotic concentrations are indicated throughout with superscripts.

Gene deletions were achieved by transferring the Kan deletion marker from donor strains in the Keio collection[69] to *E. coli* MG1655 using P1 transduction[70] (P1*vir*). For overexpression analysis, genes were expressed from ASKA library[71]

pCA24N plasmid, induced with 0.1 mM IPTG. For AcrA purification, the periplasmic portion of AcrA without its lipoprotein signal peptide[72] was cloned in pTrc99a plasmid with an N- and C-terminal His and FLAG-epitope tags, respectively. The *tolC* gene was cloned in pTrc99a, and site-directed mutagenesis was performed as described[73].

**Motility assays.** All assays used LB as the nutrient medium. For swarm assays, plates were prepared with Bacto agar for all bacteria except for *E. coli* (which requires Eiken agar; Eiken Chem. Co. Japan) at the following agar concentrations: *E. coli* (0.5%), *S. enterica* and *P. aeruginosa* (0.6%), *S. marcescens* (0.8%), and *B. subtilis* (0.7%). For *E. coli* and *S. enterica*, 0.5% glucose was included. Poured swarm plates were dried overnight at room temperature prior to use. Plates were incubated at 30 °C for *E. coli* and *S. marcescens*, and at 37 °C for all others.

Border-crossing swarm assay[12] is illustrated in Fig. 2. Four microliters of mid-log phase bacterial culture were inoculated in the left no-antibiotic chamber and allowed to dry by leaving the lid of the petri dish open for 15 min. The plates were then incubated at 30 °C or 37 °C for 24 and 16 h, respectively, which is the time it took for bacteria in control plates to colonize the entire no-antibiotic right chamber. All plates were photographed with a Canon Rebel XSI digital camera using the "bucket of light" as a light source[74].

**Preparation of dead cells and extracts.** A 10 ml culture of *E. coli* cells (0.6 OD$_{600}$) was treated for 30 min with any one of the following antibiotics: Kan$^{250}$, Gen$^{50}$, Amp$^{500}$, and Cip$^{25}$. Cells were harvested by centrifugation (12000 × *g*, 3 min, 4 °C). The pellet was once washed with sterile HEPES buffer (1.2 mM CaCl$_2$, 1.2 mM MgCl$_2$, 2.4 mM K$_2$HPO$_4$, 20 mM HEPES, 115 mM NaCl, pH of 7.4) and resuspended in the same buffer. Efficiency of killing was monitored by CFU counts on LB agar. Fifty microliters of the dead (i.e., no detectable CFUs) cell suspension was used for the border-crossing assay. To prepare extracts from the killed cells, the protocol above was scaled up to a 50 ml culture, the final cell pellet resuspended in 1 ml of HEPES buffer, and the cell suspension lysed by passing 3 times through a SLM AMINCO French press (1200 psig units). The lysed extract was precipitated by centrifugation (12000 × *g*, 5 min, 4 °C), and the supernatant used for further purification.

Enzyme treatments of cell extracts: 100 μL were treated with 10 μl of one of the following for 30 min at 37 °C: DNaseI (Thermo, 10 μg/ml), RNase A (Thermo, 10 μg/ml), or Proteinase K (Sigma, 10 μg/ml).

**Identification of the necrosignaling factor.** The border-crossing assay was used to detect activity at all steps.

*1. Cell extract preparation*: Four litre cultures (0.6 OD$_{600}$) of *E. coli/S. enterica* cells were treated with Kan$^{250}$ for 30 min. The cells were harvested by centrifugation (5000 × *g*, 10 min, 4 °C). The pellet was washed once with HEPES buffer (1.2 mM CaCl$_2$, 1.2 mM MgCl$_2$, 2.4 mM K$_2$HPO$_4$, 20 mM HEPES, 115 mM NaCl, one SIGMAFAST™ protease inhibitor tablet, pH of 7.4). The cell pellet (3.8 g) was resuspended in 15 ml of HEPES buffer and then lysed by six passages through a French Press (1200 psig units). The resultant extract was centrifuged (10000 × *g* for 10 min at 4 °C) to collect the supernatant (50–70 mg/ml protein). 2. *Ammonium sulfate precipitation*: the supernatant was treated with ammonium sulfate (AS) (30% for *E. coli* and 35% for *S. enterica*) overnight (O/N) in a cold

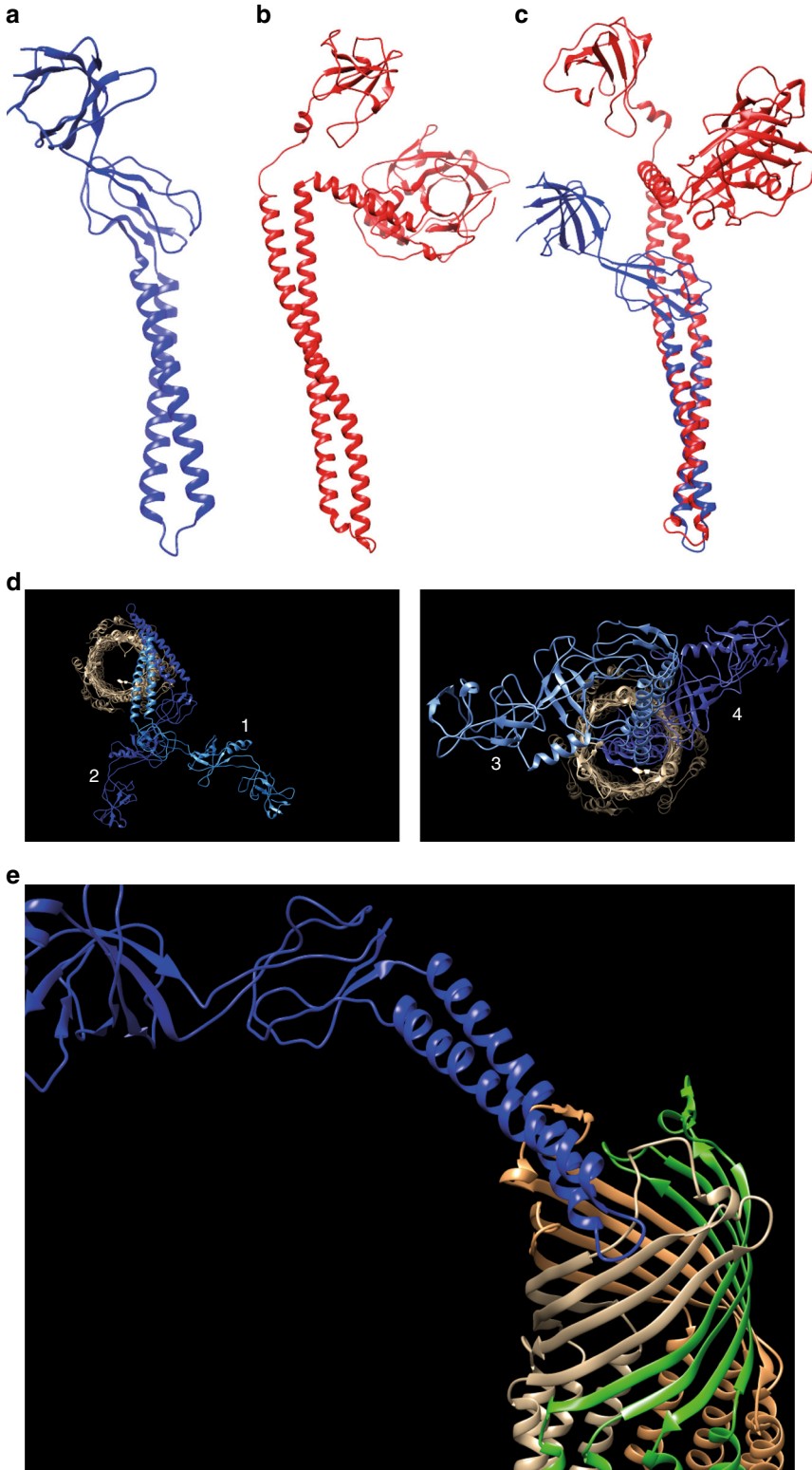

**Fig. 7 AcrA structural analysis. a–c** Structural superposition of AcrA and Colicin E3. **a** AcrA (PDB ID: 5NG5 [https://doi.org/10.2210/pdb5NG5/pdb]). **b** Colicin E3 (PDB ID: 2B5U [https://doi.org/10.2210/pdb2B5U/pdb]). **c** Superposition of **a**, **b** using Chimera matchmaker[93]. The RMSD value was 0.85 Å. **d**, **e** Model for AcrA-TolC binding. AcrA and TolC structures (PDB ID 5NG5) were used to predict possible modes of their interaction (see "Methods"). **d** The best models are presented in decreasing order (1–4) of probability. AcrA, all shades of Blue; TolC, gray, brown, and green. **e** Sideview of the best model (#1). It is physically not very probable for a full-length AcrA (~49 Å diameter) to enter the cell through the TolC pump (~29 Å diameter).

room (4 °C), while keeping it in a tumbling mode using a Barnstread thermolyne Labquake™ rotisserie shaker. The precipitate was collected by centrifugation (12000 × g for 30 min at 4 °C), the pellet was resuspended in 5 ml of HEPES buffer and dialysed (O/N) using HEPES as the exchange buffer (~5 mg/ml protein). Fifty microliters each of initial cell extract and the AS pellet were used to test for activity. 3. *Q sepharose anion exchange chromatography*: 3.5 ml of the AS fraction was loaded on a Q sepharose anion exchange chromatography column (10 cm × 0.8 cm² = 8 ml), and after the flow-through was collected, the column was washed with HEPES buffer (1 column volume). The proteins were manually eluted (0.5 ml/min) using a 50–500 mM NaCl gradient in a total of fifteen-1 ml fractions (~2 column volumes). Fractions 5 and 6 showed activity for both strains. Their protein concentrations were ~1.5 mg/ml. 4. *CM cation chromatography*: the active fractions from step 3 were pooled and loaded onto a CM cation chromatographic column (7.5 cm × 0.8 cm² = 6 ml). The flow-through was collected, followed up a 1-column volume wash with Phosphate-buffered saline (NaCl 137 mM, KCl 2.7 mM, $Na_2HPO_4$ 10 mM, $KH2PO4$ 1.8 mM, pH 7.2). The proteins were manually eluted (0.5 ml/min) with a NaCl gradient of 50–250 mM using ~2 column volumes (9 fractions × 1.5 ml each = 13.5 ml total). The first four fractions (50–125 mM NaCl) showed activity, and had protein concentrations in the range of 0.39–0.1 mg/ml.

**Mass spectrometric analysis**. Active fractions from the CM cation step were used to perform ESI-MS in a Thermo Orbitrap Fusion hybrid mass spectrometer. The proteins were Trypsin digested on the column before Tandem mass spectra were extracted. All MS/MS samples were analyzed using Sequest (Thermo Fisher Scientific; version IseNode in Proteome Discoverer 1.4.1.14) and X! Tandem (CYCLONE 2010.12.01.1). Sequest and X! Tandem were searched with a fragment ion mass tolerance of 0.80 Da and a parent ion tolerance of 10.0 PPM. Carbamidomethyl of cysteine was specified in Sequest and X! Tandem as a fixed modification. Glu->pyro-Glu of the N-terminus, ammonia-loss of the N-terminus, Gln->pyro-Glu of the N-terminus and oxidation of methionine were specified in X! Tandem as variable modifications. Oxidation of methionine was specified in Sequest as a variable modification.

Scaffold (version Scaffold_4.8.4, Proteome Software Inc., Portland, OR) was used to validate MS/MS based peptide and protein identifications. Peptide identifications were accepted if they could be established at greater than 16.0% probability to achieve an FDR less than 1.0% by the Peptide Prophet algorithm[75] with Scaffold delta-mass correction. Protein identifications were accepted if they could be established at greater than 95.0% probability to achieve an FDR less than 5.0% and contained at least four identified peptides. Protein probabilities were assigned by the Protein Prophet algorithm[76]. Proteins that contained similar peptides and could not be differentiated based on MS/MS analysis alone, were grouped for parsimony.

**Purification of AcrA**. BL21(DE3) harboring the His-AcrA-FLAG construct described above (referred to as simply AcrA henceforth) was cultured overnight in LB containing ampicillin[100]. An overnight culture was diluted 1:100 into 1 l of fresh LB, and incubated at 37 °C till 0.6 OD₆₀₀ was reached. Protein Expression was induced with the addition of 1 mM IPTG for 4 h. The cells were harvested by centrifugation, washed, resuspended in buffer [10 mM Tris-HCl (pH 8.0), 100 mM NaCl, one SIGMAFAST™ protease inhibitor tablet, 1 mg/ml lysozyme, 1 mM β-ME, 50 mM imidazole], and sonicated on ice. The supernatant was collected after centrifugation (12000 × g, 30 min, 4 °C). Protein was purified using HisTrap HP (1 ml) column (GE) according to the manufacturer's protocol with slight modifications. A 5-step gradient of imidazole (100, 200, 300, 400, and 500 mM) was used to elute the bound protein manually (2 column volumes for each step) and 1 ml fractions were collected. The fractions were dialyzed against two changes of buffer A (10 mM Tris-HCl (pH 8.0), 100 mM NaCl, 1 mM EDTA) overnight. All fractions were active.

**RNA-seq analysis**. *E. coli* cells were grown in 10 ml broth (planktonic) culture (0.6 OD₆₀₀) in absence of antibiotic and harvested with or without the treatment of antibiotic (see Figure legends for specific experiments). *E. coli* swarm cells from the right chamber in the border-crossing assay (with or without antibiotics) were resuspended in 3 ml LB medium and harvested. The harvested cells were resuspended in 1 ml ice-chilled LB medium, kept on ice for 2 min, and pelleted by centrifugation (4 °C, 10000 × g, 2 min). Every experimental condition tested had two biological replicates. Total RNA was isolated from these samples using Qiagen RNeasy® Protect Bacteria Mini Kit and following the enzymatic lysis method. The RNA was then used for library preparation using NEBNext Ultra II Library Prep Kit and sequenced on an Illumina NextSeq 500 platform (SR 75) yielding a total of 267.1 million reads for 10 samples. Sequence quality was determined by FastQC v0.13 and MultiQC[77]. The raw files were processed (Cutadapt[78]), aligned (Bowtie[79]), mapped (Samtools[80], Bedtools[81]) to reference genome (GenBank ID U00096.3), normalized and analyzed (DESeq2[82]), and visualized (R studio, Microsoft Excel, and Graphpad). The gene enrichment analysis was performed by using DAVID[83,84].

**Microscopy**. The Kan marker from ΔtolC and ΔacrA strains were removed using a pCP20 based method[85] to generate ΔtolC^ΔK and ΔacrA^ΔK respectively. To prepare

the AcrA probe, DYKDDDDK Tag mouse monoclonal antibody (FG4R) (Thermo Fisher) was labeled with a quantum dot (Qdot 705) using a click chemistry kit (SiteClick™ Qdot™ 705 Antibody Labeling Kit, Thermo Fisher)[86]. 100 μl of either *E. coli* planktonic culture in LB medium (0.6 OD₆₀₀, treated with Kan[25] for 30 min) or swarm cells collected from border-crossing assay plates (Kan[25], as described in RNA-seq section) were incubated with membrane dye FM1-43 (0.1 mg/ml, Film-Tracer™ FM™ 1–43, Thermo Fisher) for 15 min at 37 °C in the dark. 100 ng of purified AcrA was added to these cell suspensions and incubated for 15 more min. The cells were then pelleted, washed, resuspended in 100 μl of 1× PBS (pH 7.4), and Qdot 705-labeled antibody was added (1:1000) for 15 min incubation at room temperature. The excess antibody was removed by washing the cells by PBS buffer. Cells were then visualized using a light microscope (BX53F; Olympus, Tokyo, Japan) and cellSens software (v1.6) with minimal adjustments and alignments in Adobe Photoshop. FM1–43 was visualized using a GFP filter (excitation wavelength, 460–480 nm; emission 495–540 nm). To reduce quenching and avoid bleed-through because of overlapping excitation or emission wavelengths, the Qdot705 was visualized using a bandpass filter AT-Qdot 705 filter (Chroma, 39018) that allows excitation wavelength of 400–450 nm (425 CWL) and emission wavelength of 685–725 nm (705 CWL). After incubation with AcrA for the microscopy experiments, a subset of the swarm cell sample (200 μl) was used to determine the localization of extracellularly added AcrA[87] with modifications. The cells were pelleted with gentle centrifugation (1500 × g, 10 min, 4 °C) and resuspended in protease buffer (50 mM Tris, 7.5 mM CaCl₂, pH 8.8). The suspension was divided into two equal halves. One half was lysed with sonication (three cycles of 30 s on and 30 s off) and supernatant was collected. The un-lysed cells and the supernatant of the lysed sample were incubated with trypsin (50 μg/ml, Thermo Fisher) for 30 min at RT. The protease was then inactivated with 0.1 M PMSF. These samples were used for western blot analysis with DnaK as cytoplasmic control using mouse monoclonal anti-DnaK antibody (abcam).

**Efflux pump assays**. Planktonic and swarm cells were harvested as for RNA-seq analysis. Nile Red and Alamar Blue assays were performed as described[29] with modifications. Each sample (1 ml) was centrifuged for 10 min at 5000 × g; the pellet was suspended in potassium phosphate buffer (PPB, 20 mM potassium phosphate, 1 mM MgCl₂, pH 7.0) to obtain OD₆₀₀ of 1.0.

Nile Red assay. Samples were incubated with 10 mM CCCP (prepared in 50% DMSO) for 30 min at RT followed by addition of 10 mM Nile Red (VWR) at 37 °C and shaken at 200 rpm for 30 min. These cells were then kept in room temperature for 15 min without shaking, harvested via centrifugation, and supernatant was carefully and completely discarded. The resultant pellet was resuspended in PPB to obtain OD₆₀₀ of ~1.0; a 10× dilution of that suspension was transferred to a quartz cuvette to measure fluorescence (excitation at 552 and emission at 636 nm) using a QuantaMaster spectrofluorometer every 10 s for 100 s with intermittent manual stirring. The efflux of Nile Red was triggered by addition of 100 μl of 1 M glucose followed by measuring fluorescence every 10 s for 200 s. In a subset of samples, 100 ng of purified AcrA was added along with the glucose. The obtained absorbance ($A_{636}$) was normalized with OD₆₀₀.

Alamar Blue assay. 20 μl of Alamar Blue liquid reagent (Invitrogen) was added to 180 μl of cell suspension. The fluorescence of the samples was measured (excitation at 565 nm, emission at 590 nm) every 5 min for 30 min followed by every 10 min for 30 more min. In a subset of samples, 20 ng of purified AcrA was added before the addition of Alamar Blue.

Fleroxacin assay. Planktonic and swarm cells were harvested as before. A cell suspension of ~0.6 OD₆₀₀ was prepared in 2 ml LB and each sample was divided into two equal halves. Appropriate amounts of Fleroxacin (VWR) were added to one half, and the cells were incubated for 30 min while shaking at 37 °C[31]. The other half was used as the untreated control (990 μl) and to calculate CFU (10 μl). The cells were pelleted, washed once, resuspended in 1 ml of 50 mM sodium phosphate buffer (pH 7.2), and sonicated to lyse (3 rounds of 30 s on and 30 s off). The supernatant was used to determine fluorescence at 442 nm with excitation at 282 nm. The actual Fleroxacin fluorescence was calculated by subtracting the fluorescence of the untreated sample from fluorescence of treated ones. The concentration of Fleroxacin was determined by comparing with a standard curve of known Fleroxacin concentrations.

Disc-diffusion assay. Swarm cells were collected from the right chamber of a border-crossing plate containing the indicated Kan concentrations in Fig. 5e, 2 h after the cells crossed the border, and planktonic cultures were incubated for 2 h in the same Kan concentrations. 100 μl of collected cells were kept at 95 °C for 15 min, chilled in 4 °C for 5 min, and centrifuged at 10,000 × g for 5 min. The supernatants were spotted on filter discs, dried for 2 h, and placed on LB plates. The supernatant was used for deposit on discs. Kan is a thermostable antibiotic[32]. The CFUs were estimated for the cultures used for cell extract preparation. The following formula was applied to calculate intracellular [Kan]:([Kan]estimated from diameter of disc)/(CFU of culture).

**Killing curves**. An O/N culture of *E. coli* was sub-cultured (1 ml) to obtain 0.6 OD₆₀₀. The swarm cells were collected from a regular swarm plate using 1 ml of LB and finally diluted to 0.6 OD₆₀₀. These cell suspensions were incubated with various concentrations of Kan for 2 h at 37 °C with shaking at 200 rpm. A fraction of cells (100 μl) were collected every 30 min and dilution plating was used for measuring CFU counts. To confirm CFU counts, another fraction of cells (100 μl) from

some samples were stained with LIVE/DEAD Bac Light Bacterial Viability Kit (Thermo Fisher) for 15 min by adding 5 μl of SYTO9 and PI dye each. The cells were visualized using a fluorescent microscope (BX53F; Olympus, Tokyo, Japan) with GFP and RFP filters for SYTO9 and PI dyes respectively.

**External AcrA docking on TolC.** AcrA and TolC structures were obtained from PDB[88] (ID: 5NG5)(rcsb.org). One AcrA molecule was docked on to TolC tripartite complex using the HADDOCK2.2 web server[89] (https://haddock.science.uu.nl/services/HADDOCK2.2/) with default parameters. The docking was guided by excluding the region of TolC that normally interacts with AcrA in the periplasm (Gln142-Thr152 and Ala360-Val372), but the whole AcrA structure was used. The resulting top four models had HADDOCK scores ranging from −172 to −139, z-scores from −2.5 to −0.4, and RMSD of 0.7 to 2.3 Å.

**AcrA sequence comparison and distribution.** AcrA sequences from five organisms studied here were used for multiple sequence alignment using Clustal Omega[90] algorithm and visualized by Jalview[91]. The sequences were obtained from NCBI protein database (https://www.ncbi.nlm.nih.gov/protein). A cladogram was generated by neighbor-joining method without distance corrections using Clustal Omega showing relatedness of these AcrA sequences.

*E. coli* AcrA and TolC sequences were used separately as inputs in the homology-based protein sequence alignment algorithm phmmer in HMMER[92] web server (https://www.ebi.ac.uk/Tools/hmmer/) with a significant *e* value and hit cutoffs of $10^{-20}$ and 0.03, respectively. The hits were then visualized in HMMER taxonomy, and counts for each taxon in the Bacterial domain were used for generating a heat map (Supplementary Fig. 11).

**Modeling bacterial response to antibiotics.** The percent survival of the bacterial population for a given antibiotic concentration as a function of time was modeled as a set of ordinary differential equations. Initially, rate of change in a specific bacterial population was modeled as simple first-order reaction given by Eqs. 1–3.

$$A \xrightarrow{k_a} D, \tag{1}$$

$$\frac{dA_n}{dt} = -k_a[A]_n, \tag{2}$$

$$[A]_{n+1} = [A]_n - k_a[A]_n. \tag{3}$$

The population live cells, A, will convert into dead cells, D, with some rate of change, $k_A$. Here we have simplified all the processes that lead to changes in the bacterial population to a single term $k_a$ that ideally represents a convolution of both bacterial growth and death. Time-step updates to the population of A cells is given by Eq. 3. The percent survival for a single population of cells is given by Eq. 4.

$$S_n = [A]_n/[A]_0. \tag{4}$$

The percent survival of cells, $S_n$, is given by dividing the remaining cells at any time-point n, $[A]_n$, by the initial starting population, $[A]_0$. A Gibbs sampler was built to search for suitable parameters for the rate of change for the population based on reducing the mean-square error (MSE) between the simulation and experiment over 100,000 iterative steps. Simulations for a homogenous population exhibited deviation between experiment and simulation results as measured by MSE for swarming cells ($10^{-3}$–$10^{-2}$) and significantly better results for planktonic cells ($10^{-6}$–$10^{-3}$) (Fig. 1a).

The failure of the simulation to reproduce the experiment was hypothesized to be the result of the simplicity of the starting assumption that there is one population of cells with a singular response rate to the antibiotics. A more complex approach was developed using a heterogeneous starting population composed of two subpopulations (A and B) that follow the same first-order kinetics shown in Eqs. 1–3 but are allowed to differ in rates of change and relative starting population to one another. The survival percentages were calculated using Eq. 5.

$$S_n = \frac{[A]_n + [B]_n}{[A]_0 + [B]_0}. \tag{5}$$

Using the Gibbs sampler to explore the independent rates for both subpopulations, the simulation was much better able to replicate the experimental results for swarming cells with an MSE range of $10^{-7}$–$10^{-5}$ typically conferring a hundred to ten-thousand fold better fit over a homogenous simulation (Fig. 1a). The differences between the populations usually converged into a solution where cells of one population were 10–100 times faster at dying, and are referred to as the fast-dying (FD) cells. Typically, the slow-dying cells (SD) where twice as numerous in starting populations than the FD cells (50–70%). In addition, magnitude of the difference of rates of death, $|k_A - k_b|$, between the FD and SD cells decreased as a function of antibiotic concentration. In contrast, simulations on heterogeneous swimming cells showed a drastically lower rate of improvement compared to homogenous cells with MSE ranges ranging $10^{-7}$–$10^{-3}$, typically indicating a 2–100-fold improvement in fit. The differences in MSE scores between the homogeneous and heterogeneous models are statistically significant for both swarming (*p* value: $6.5 \times 10^{-4}$) and planktonic cells (*p* value: $4.5 \times 10^{-3}$). Taken as aggregate, there is sufficient evidence that two populations of bacterial cells exist

within the swarming populations that differ in initial concentrations and susceptibility to kanamycin.

**Gibbs sampler.** Bacterial models where built using Eqs. (3) and (5) while determining the percent survival at discrete time steps for comparison to experimental data and MSE calculation. Each model had $2 \times N$ variables that could be altered to determine a better fit, where N is the number of subpopulations. The variables for each subpopulation are the initial size of the population ($A_0$, $B_0$, $C_0$, etc.) referred to as *starting_size* and the rate of change ($k_a$, $k_b$, $k_c$, etc.) referred to as *death_rate*. The Gibbs sampler was initiated with equal *starting_size* and random *death_rates* chosen from a uniform distribution between 0.1 and 0.001 for each subpopulation. A resulting initial MSE for the model was calculated and referred to as old_MSE. To converge on a reasonable value for each variable and reduce the MSE, the following Gibbs-sampling algorithm was employed as follows for 100,000 iterations:

(1) Choose a variable randomly. This variable is referred to as old_variable.
(2) The chosen variable selects a proposed value for the variable from a normal distribution centered on the current value of the variable with a standard deviation one-tenth the size of the current value. This value will be referred to as new_variable.
(3) Estimate the new_MSE using the new_variable instead of the old_variable. Unselected variables will remain the same.
(4) If the new MSE is lower than the old MSE:
    set old_variable = new_variable.
    Set old_MSE = new_MSE.

Variables were chosen at random for each iteration, so as not to introduce an order selection bias to determining variable selections.

**Statistics and reproducibility.** Unless mentioned otherwise, all the experiments were performed with three biological replicates.

**Reporting summary.** Further information on research design is available in the Nature Research Reporting Summary linked to this article.

## Data availability
The crystal structures of proteins with PDB IDs 5NG5 and 2B5U were obtained from publicly available PDB database (https://www.rcsb.org/). The RNA-seq data were deposited to SRA with BioProject ID PRJNA640755. Source data are provided with this paper.

## Code availability
The relevant codes for mathematical modeling are provided in the Methods.

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

## Acknowledgements

This work was supported by National Institutes of Health Grant GM118085 and in part by the Robert Welch Foundation Grant F-1811. We thank the Barrick lab here at UT Austin for the ASKA library.

## Author contributions

S.B. and R.M.H. set up the experimental design, S.B. performed the experiments, S.B. and R.M.H. wrote the paper. D.W. performed the mathematical modeling.

## Competing interests

The authors declare no competing interests.
