## [Peer Review File · Nature Communications]

Reviewers' comments:

Reviewer #1 (Remarks to the Author):

This is an interesting study that demonstrates a novel "necrosignaling" moonlighting function for AcrA released by dead cells to stimulate drug efflux in swarming cells of E. coli. This is a comprehensive study that not only demonstrates necrosignaling as a mechanism of drug resistance but also identifies the released factor and mechanism of signalling by binding to TolC. I have a few minor comments for consideration:

1. The use of acronyms SR and STRIVE interchangeably makes this manuscript difficult to follow. The STRIVE terminology is unnecessary. The abbreviation SR is also unnecessary. "Swarming resistance" would be sufficient throughout. If keeping SR then this should be defined in the Main text at first appearance.
2. Figure 1 legend- line 228- insert the word "Figure" so reads "...Figure S1A..."
3. Line 132. Insert the word "of" so that sentence reads "...we tested the effect of both deleting...."
4. Line 146- what evidence is there that FM 1-43 is specific for OM labelling? It could also be labelling the IM and due to the resolution limit of these images is not possible to distinguish where the Q-dot labelled AcrA is located on or in the cell. These images show only if there is localisation of the Qdot labelled AcrA to the bacterial cell. It is not necessary to over-interpret these images. The subsequent trypsin experiments are sufficient to demonstrate extracellular binding of AcrA.

Reviewer #2 (Remarks to the Author):

Swarming is a way of collective surface movement executed by several bacterial species. Adaptation to antibiotics can be impacted by swarming, albeit the exact molecular mechanisms remain widely elusive. In the present manuscript, the authors suggest that AcrA proteins from death cells can act as signal to trigger drug efflux in alive cells through TolC. While this finding (if true) might mark an interesting conceptual advance, I have several major comments and questions that require answers before I can recommend publication.

In supplemental Fig. 1, the authors show a live/dead screen. However, even in the Kan25 sample the green fluorescent signal seems reduced to me. I do not see a convincing difference. Moreover, the whole figures miss error bars and statements for the statistics.

In Fig. 1b, the authors show ,Dead cells'/Kan250. However, the authors never show that the cells are indeed 'dead' after the kan250 treatment. I clearly miss controls here? The authors should clarify if whether a suspension of dead cells was applied or only the supernatant. In the next experiment, the only compare supernatant + proteinase K treatment. So what was done? Otherwise, the authors would mix apples with pears.

Fig 1 c/d: Clearly misses the wildtype control!

Fig 1e. How were the dead cells killed? Original data are missing. It is hard to judge the quality of the data from what is presented!

Can the authors comment why they didn't show the influence of Crp? This protein also popped up in both tested organisms? It is in the text just stated as "data not shown".

How was AcrA purified? The authors must present the quality of their purified protein. Otherwise, it cannot be excluded that contaminants are the cause of the effects/observations.

I'm missing a list of the used strains and plasmids – what were the antibiotic resistances of these? Especially, the tolC mutant? The authors must state that – it is hard to judge the quality,

In order to make the claim that AcrA is interacting with TolC, the authors must show a direct binding. This can easily be done by an in vitro approach.

Figure S8. Again wild type control is missing! Why do some plates show inhibition of swarming at low Kan concentrations, but weaker at higher concentrations? e. g. Arg55Leu or N100

Fig 2. Were the protein expression levels of the protein mutants the same? Please show a quantification (e.g. by western blotting).

Fig. S9 Why are planktonic cells much smaller? Is this a general feature? Please comment.

Figs. 2d/e . Again controls are missing! Either show Qdot705 without AcrA or AcrA without FLAG tag. In other words, can you exclude that Qdot705 is non-specifically interacting with the cells and/ or TolC?

Fig. S12. Please provide the raw data.

How can the claimed results be connected to chemotherapy-resistant cancer? That seems very speculative to me! At least remove that claim from the abstract.

Reviewer #3 (Remarks to the Author):

The manuscript describes a mechanism for the adaptive antibiotic resistance in swarming cells (called SR) of (mostly) *Escherichia coli*. Specifically, the authors demonstrate that cell death in a sub-population of swarming bacteria leads to the release of a "necrosignal" that increases survival of the remaining cells in presence of lethal and elevated concentrations of antibiotics. The authors identify AcrA as the necrosignal released from dead cells and show that it binds to TolC to mediate SR. They identify the region of AcrA required for this function, show that tagged AcrA binds to the outer membranes of cells in a manner that depends on the presence of TolC and of specific residues on TolC. Last they also show that swarming cells are pre-programmed to be less permeable, have increased expression of efflux pumps and ROS pathways, which could contribute to increased antibiotic resistance.

The manuscript is generally well written and it provides a compelling set of interesting data. However, there are instances where the experimental design is not clear and where some interpretations should be either modified, clarified or re-evaluated.

1. The authors use the term "necrosignal" for the role of AcrA but whether it is acting as a signal seems to be true only for swarming cells and perhaps also, only for some antibiotics. First, unless I am mistaking, it appears that the antibiotics used here are effluxes from the cells using the AcrAB/D-TolC efflux pump. This begs the question of whether the role of AcrA as a "necrosignal" applies to antibiotics that would not be effluxed in the same way and whether this is a very specific effect of AcrA that reveals some basic structure relationship with respect to its interaction with TolC. The structural homology with Colicin E3 is intriguing in that respect.

2. Related to the above comment, the authors show that addition of AcrA to swarm cells induced the expression of a overlapping subset of genes which those which expression is modulated by the addition of antibiotics. However, the effect of adding AcrA to planktonic cells on these genes is not tested and in fact, per Fig S7, the addition of AcrA to planktonic cells only promotes a small increase in survival of cells treated with Isn concentrations near MIC99. If AcrA is to be a "necrosignal", wouldn't it be expected to also function on planktonic cells? As the authors state, (lines 126-127), the result in Fig. S7 suggest that the physiology of swarming cells play an important role (which they go on to investigate in the next paragraph). In light of the findings of the interaction with TolC and the presence of tagged AcrA on the OM, wouldn't this suggest that the "necro-signal" is in fact not AcrA but the pair AcrA-TolC?

3. The binding of AcrA to TolC is suggested by a combination of microscopy and enzymatic treatment of cells with trypsin, with cells also expressing mutated versions of TolC. The contrast (or size) of images shown in Fig 2 should be increased. A scale bar should also be added to these figures. What was the resolution? Any possibility to have an idea of the stoichiometry of binding?

4. The conclusion that release of AcrA by dead cells and its binding to TolC on live cells to instantly stimulate efflux is not a straightforward conclusion that can be drawn from the data shown in Fig. 2d and 3b. The data may be consistent with this model but there could also be alternative explanations such as that could be missed with the resolution and methods used here- perhaps the binding is very weak and only (larger) cells that express more TolC are seen with binding to AcrA. How far would AcrA diffuse before being proteolytically cleaved? Is AcrA released by cells which have not completely lysed yet? These and perhaps many other considerations should be used to generate more circumspect conclusions.

5. The results on ROS activation and the effect of hydrogen peroxide-mediated killing of cells are not discussed much but the authors emphasize them in the discussion section. Should a few comments be added here? Similarly, the structural homology of AcrA with colicin B3 is intriguing and only appears as a sentence in the discussion.

6. Minor comments:

Line 431- delete ", " after "section"

Line 462- what is PPB?

Reviewer #1:

This is an interesting study that demonstrates a novel “necrosignaling” moonlighting function for AcrA released by dead cells to stimulate drug efflux in swarming cells of *E. coli*. This is a comprehensive study that not only demonstrates necrosignaling as a mechanism of drug resistance but also identifies the released factor and mechanism of signalling by binding to TolC. I have a few minor comments for consideration:

1. The use of acronyms SR and STRIVE interchangeably makes this manuscript difficult to follow. The STRIVE terminology is unnecessary. The abbreviation SR is also unnecessary. “Swarming resistance” would be sufficient throughout. If keeping SR then this should be defined in the Main text at first appearance.

- STRIVE is removed. We have kept SR and defined it in the Main text at first appearance.

2. Figure 1 legend- line 228- insert the word “Figure” so reads “...Figure S1A...”

- Fixed.

3. Line 132. Insert the word “of” so that sentence reads “..we tested the effect of both deleting....”

- Fixed

4. Line 146- what evidence is there that FM 1-43 is specific for OM labelling? It could also be labelling the IM and due to the resolution limit of these images is not possible to distinguish where the Q-dot labelled AcrA is located on or in the cell. These images show only if there is localisation of the Qdot labelled AcrA to the bacterial cell. It is not necessary to over-interpret these images. The subsequent trypsin experiments are sufficient to demonstrate extracellular binding of AcrA.

- Agreed. The text has been updated accordingly (lines 145-151).

Reviewer #2:

Swarming is a way of collective surface movement executed by several bacterial species. Adaptation to antibiotics can be impacted by swarming, albeit the exact molecular mechanisms remain widely elusive. In the present manuscript, the authors suggest that AcrA proteins from death cells can act as signal to trigger drug efflux in alive cells through TolC. While this finding (if true) might mark an interesting conceptual advance, I have several major comments and questions that require answers before I can recommend publication.

In supplemental Fig. 1, the authors show a live/dead screen. However, even in the Kan25 sample the green fluorescent signal seems reduced to me. I do not see a convincing difference. Moreover, the whole figures miss error bars and statements for the statistics.

- The relevant data in this figure are the green and red cell counts, not their color intensity. Error bars have been added to Figure S1a, and statistical tests are provided in Table S2.

In Fig. 1b, the authors show ,Dead cells'/Kan250. However, the authors never show that the cells are indeed 'dead' after the kan250 treatment. I clearly miss controls here? The authors should clarify of whether a suspension of dead cells was applied or only the supernatant. In the next experiment, the only compare supernatant + proteinase K treatment. So what was done? Otherwise, the authors would mix apples with pears.

- We state under Methods that the “Efficiency of killing was monitored by CFU counts on LB agar”. Dead = no detectable CFUs on LB agar. A clarifying phrase is now added here (lines 359-360). The legend to Fig. 1b clearly distinguishes use of dead cells vs the dead cell extracts, as does the text (lines 92-93 and 99-100). The effect of the supernatant alone was shown in Figure S3b (middle plate); this is now indicated in Fig. 1b legend.

Fig 1 c/d: Clearly misses the wildtype control!

- The WT control is found in Fig 1, b2. This is now added to Fig. 1c legend.

Fig 1e. How were the dead cells killed? Original data are missing. It is hard to judge the quality of the data from what is presented!

- Two different Kan concentrations were required for complete killing, depending on the organism. The killing method was indicated in Methods (lines 350-352); the legend now points the reader there. The raw plate data have been added as a new 'd' panel to Figure S3.

Can the authors comment why they didn't show the influence of Crp? This protein also popped up in both tested organisms? It is in the text just stated as “data not shown”.

- Deletion of *crp* severely represses swarming¹. Killed cells overexpressing *crp* in WT did not enhance resistance to Kan⁷⁰, similar to *uspE* and *yhdC* shown in Fig. 1d. Including this negative *crp* plate in d without its deletion partner in c would have detracted from the balance of the images in panels b-d. For these two reasons, the negative *crp* result was not shown in d.

How was AcrA purified? The authors must present the quality o their purified protein. Otherwise, it cannot be excluded that contaminants are the cause of the effects/observations.

- The purification results are shown in a panel 'd' added to Fig. S5.

I'm missing a list of the used strains and plasmids – what were the antibiotic resistances of these? Especially, the *tolC* mutant? They authors must state that – it is hard to judge the quality,

- The strain and plasmid list are now provided in Table S1.

In order to make the claim that AcrA is interacting with TolC, the authors must show a direct binding. This can easily be done by an *in vitro* approach.

- This experiment is actually not easy, and not possible at this time. Please take a look at Fig. 1B. The established interaction of AcrA is with the periplasmic face of TolC. The results reported in our paper show AcrA interaction with the OM face of TolC. Unless there is a way to generate only the OM face, it is not possible to distinguish the two *in vitro*. TolC is a trimeric assembly of an α - β barrel protein, in which each monomer weaves in and out between OM β -sheets and periplasmic α -helices². The assembly pathway of this complex is unique³ in that monomeric polypeptides translocate through the inner membrane and are released in the periplasm, where they mature and possibly oligomerize before insertion in the OM. A characteristic proteinase K-resistant fragment generated by cleavage at a single, periplasmically exposed, protease-sensitive site of the membrane-anchored trimer signals completion of assembly. Thus, truncated variants of TolC encoding only the β -sheets are not expected to assemble either *in vitro* or in the OM.

We believe Fig. 2 provides convincing genetic and microscopy evidence of AcrA interaction with the OM face of TolC.

Figure S8. Again wild type control is missing! Why do some plates show inhibition of swarming at low Kan concentrations, but weaker at higher concentrations? e. g. Arg55Leu or N100

- The WT control for TolC is found in Figure 2, a2. This note is now added to Fig. S8 legend. An AcrA control has been added to this panel. The issues with images such as Arg55^{Ala} or Δ N100 are ones of image brightness/contrast and figure compression. (Note: The Arg55 substitution was to Ala not Leu. This was reported correctly in Fig. 2e, but incorrectly in Fig. 2c. This is now corrected). If you zoom the image 4X, you will see there is no inconsistency. For example, you will see for Arg55^{Ala} that there is no SR beyond Kan³⁰, i.e. no cross-over and growth on any Kan concentration over Kan³⁰. Similarly, for Δ N100, there is no SR beyond Kan⁴⁰.

Fig 2. Were the protein expression levels of the protein mutants the same? Please show a quantification (e.g. by western blotting).

- The TolC mutants used in this study have already been reported to be physiologically functional without any major changes in their expression as seen in western blots (see Table 1 in ref⁴). We know that these mutants are functional in our hands because Δ *tolC* is \sim 3/4^h as proficient at swarming as the WT, and all our TolC mutants restored swarming to WT when expressed in the Δ *tolC* strain. With respect to the AcrA mutants, most deletions had no effect i.e. retained functionality, until the deletion extended into

the HTH region. When only this region was expressed, it worked as well as the whole protein in promoting SR (see Fig. 2c *acrA*Δ72ΔN75).

Fig. S9 Why are planktonic cells much smaller? Is this a general feature? Please comment.

- Swarm cells of *E. coli* tend to be slightly longer than planktonic cells⁵. This comment is added to the legend.

Figs. 2d/e . Again controls are missing! Either show Qdot705 without AcrA or AcrA without FLAG tag. In other words, can you exclude that Qdot705 is non-specifically interacting with the cells and/ or TolC?

- This control is inherent in the observation in Fig. 2e, where there is no binding of Qdot705 to Δ*tolC* or the R55A mutant. Nevertheless, the Qdot control has been added to Figure S9d.

Fig. S12. Please provide the raw data.

- Provided.

How can the claimed results be connected to chemotherapy-resistant cancer? That seems very speculative to me! At least remove that claim from the abstract.

- Removed. The connection to chemotherapy-resistant cancer is that while the chemicals kill cancer cells, these cells are also engaged in drug efflux.

Reviewer #3 (Remarks to the Author):

The manuscript describes a mechanism for the adaptive antibiotic resistance in swarming cells (called SR) of (mostly) *Escherichia coli*. Specifically, the authors demonstrate that cell death in a sub-population of swarming bacteria leads to the release of a "necrosignal" that increases survival of the remaining cells in presence of lethal and elevated concentrations of antibiotics. The authors identify AcrA as the necrosignal released from dead cells and show that it binds to TolC to mediate SR. They identify the region of AcrA required for this function, show that tagged AcrA binds to the outer membranes of cells in a manner that depends on the presence of TolC and of specific residues on TolC. Last they also show that swarming cells are pre-programmed to be less permeable, have increased expression of efflux pumps and ROS pathways, which could contribute to increased antibiotic resistance.

The manuscript is generally well written and it provides a compelling set of interesting data. However, there are instances where the experimental design is not clear and where some interpretations should be either modified, clarified or re-evaluated.

1. The authors use the term "necrosignal" for the role of AcrA but whether it is acting as a signal seems to be true only for swarming cells and perhaps also, only for some antibiotics. First, unless I am mistaking, it appears that the antibiotics used here are effluxed from the cells using the AcrAB/D-TolC efflux pump. This begs the question of whether the role of AcrA as a "necrosignal" applies to antibiotics that would not be effluxed in the same way and whether this is a very specific effect of AcrA that reveals some basic structure relationship with respect to its interaction with TolC. The structural homology with Colicin E3 is intriguing in that respect.

- Yes, necrosignaling is largely a swarming-specific SR response; planktonic cells are only weakly responsive.

Fig. S3a shows SR to Ampicillin, Ciprofloxacin, and Gentamycin. While the experiments in this work used antibiotics that are effluxed through the AcrAB/D-TolC pumps as well as other RND pumps^{6,7}, previous studies have also found SR to antibiotics such as polymixin⁸, nalidixic acid, piperacillin, tobramycin, trimethoprim⁹, etc, which are not limited to efflux through AcrAB/D-TolC⁷. In addition, the RNA seq data shows upregulation of all other classes of efflux pumps (Fig 3a and 4). We would argue that a necrosignaling module would elicit a general response to multiple classes of antibiotics. The structural homology to TolC is indeed intriguing. A speculative idea is that toxin-induced death might give rise to competition of both AcrA and the toxin for extracellular binding to TolC, increasing the chances of survival within a population.

2. Related to the above comment, the authors show that addition of AcrA to swarm cells induced the expression of a overlapping subset of genes which those which expression is modulated by the addition of antibiotics. However, the effect of adding AcrA to planktonic cells on these genes is not tested and in fact, per Fig S7, the addition of AcrA to planktonic cells only promotes a small increase in survival of cells treated with Isn concentrations near MIC99. If AcrA is to be a "necrosignal", wouldn't it be expected to also function on planktonic cells? As the authors state, (lines 126-127), the result in Fig. S7 suggest that the physiology of swarming cells play an important role (which they go on to investigate in the next paragraph). In light of the findings of the interaction with TolC and the presence of tagged AcrA on the OM, wouldn't this suggest that the "necro-signal" is in fact not AcrA but the pair AcrA-TolC?

Well, the 'signal' needs a 'receptor'. AcrA-TolC are a signal-receptor pair. Swarm cells have increased numbers of both (Fig. 3a), which might explain why necrosignaling is largely a swarming-specific phenomenon. There are also widespread differences in gene expression profiles between swarm and planktonic cells, as well as different metabolic states¹⁰, which might play into the specificity of the necrosignaling response.

3. The binding of AcrA to TolC is suggested by a combination of microscopy and enzymatic treatment of cells with trypsin, with cells also expressing mutated versions of TolC. The contrast (or size) of images shown in Fig 2 should be increased. A scale bar should also be added to these figures. What was the resolution? Any possibility to have an idea of the stoichiometry of binding?

- The contrast has been adjusted and scale bar has been added. The resolution of the microscope used was 0.1 μM . We do not have hard evidence of stoichiometry, but from comparing minimum numbers of AcrA molecules/cell required for SR at Kan⁵⁰ (~170, in Figure 1f) and reported numbers of TolC molecules/planktonic cell (~885, from Ecocyc database that adapted from ¹¹), we hypothesize that multiple AcrA molecules could bind to one TolC pump.

4. The conclusion that release of AcrA by dead cells and its binding to TolC on live cells to instantly stimulate efflux is not a straightforward conclusion that can be drawn from the data shown in Fig. 2d and 3b. The data may be consistent with this model but there could also be alternative explanations such as that could be missed with the resolution and methods used here- perhaps the binding is very weak and only (larger) cells that express more TolC are seen with binding to AcrA. How far would AcrA diffuse before being proteolytically cleaved? Is AcrA released by cells which have not completely lysed yet? These and perhaps many other considerations should be used to generate more circumspect conclusions.

- The 'instant' inference is derived from Nile Red efflux data in Fig. 3b. While we agree that this picture may be derived from a subset of cells that bind AcrA well, it doesn't detract from the conclusion that AcrA is capable of eliciting a fast response in the population. We have added the phrase 'efflux in the population' in the text (line 190). While we have no data to address the other two questions, this knowledge will not detract from the overall response seen in Fig. 3b.

5. The results on ROS activation and the effect of hydrogen peroxide-mediated killing of cells are not discussed much but the authors emphasize them in the discussion section. Should a few comments be added here? Similarly, the structural homology of AcrA with colicin B3 is intriguing and only appears as a sentence in the discussion.

- The main text now includes a discussion on ROS (lines 176-184). A speculative scenario regarding the AcrA-colicin structural homology is added to the Discussion (lines 202-206).

6. Minor comments:

Line 431- delete ", " after "section"

- Fixed

Line 462- what is PPB?

- PPB stands for potassium phosphate buffer (added).

1 Inoue, T. *et al.* Genome-wide screening of genes required for swarming motility in *Escherichia coli* K-12. *Journal of bacteriology* **189**, 950-957, doi:10.1128/JB.01294-06 (2007).

2 Touze, T. *et al.* Interactions underlying assembly of the *Escherichia coli* AcrAB-TolC multidrug efflux system. *Molecular microbiology* **53**, 697-706, doi:10.1111/j.1365-2958.2004.04158.x (2004).

3 Tikhonova, E. B., Yamada, Y. & Zgurskaya, H. I. Sequential mechanism of assembly of multidrug efflux pump AcrAB-TolC. *Chemistry & biology* **18**, 454-463, doi:10.1016/j.chembiol.2011.02.011 (2011).

- 4 German, G. J. & Misra, R. The TolC protein of *Escherichia coli* serves as a cell-surface receptor for the newly characterized TLS bacteriophage. *Journal of molecular biology* **308**, 579-585, doi:10.1006/jmbi.2001.4578 (2001).
- 5 Partridge, J. D. & Harshey, R. M. More than motility: *Salmonella* flagella contribute to overriding friction and facilitating colony hydration during swarming. *Journal of bacteriology* **195**, 919-929, doi:10.1128/JB.02064-12 (2013).
- 6 Sulavik, M. C. *et al.* Antibiotic susceptibility profiles of *Escherichia coli* strains lacking multidrug efflux pump genes. *Antimicrobial agents and chemotherapy* **45**, 1126-1136, doi:10.1128/AAC.45.4.1126-1136.2001 (2001).
- 7 Yilmaz, C. & Ozcengiz, G. Antibiotics: Pharmacokinetics, toxicity, resistance and multidrug efflux pumps. *Biochemical pharmacology* **133**, 43-62, doi:10.1016/j.bcp.2016.10.005 (2017).
- 8 Butler, M. T., Wang, Q. & Harshey, R. M. Cell density and mobility protect swarming bacteria against antibiotics. *Proceedings of the National Academy of Sciences of the United States of America* **107**, 3776-3781, doi:10.1073/pnas.0910934107 (2010).
- 9 Lai, S., Tremblay, J. & Deziel, E. Swarming motility: a multicellular behaviour conferring antimicrobial resistance. *Environmental microbiology* **11**, 126-136, doi:10.1111/j.1462-2920.2008.01747.x (2009).
- 10 Kim, W. & Surette, M. G. Metabolic differentiation in actively swarming *Salmonella*. *Molecular microbiology* **54**, 702-714, doi:10.1111/j.1365-2958.2004.04295.x (2004).
- 11 Ishihama, Y. *et al.* Protein abundance profiling of the *Escherichia coli* cytosol. *BMC genomics* **9**, 102, doi:10.1186/1471-2164-9-102 (2008).

REVIEWERS' COMMENTS:

Reviewer #2 (Remarks to the Author):

The authors have sufficiently addressed my concerns.

Reviewer #3 (Remarks to the Author):

The revised manuscript provides additional information regarding experimental design and clarifies several of the interpretations.